# A widely conserved protein Rof inhibits transcription termination factor Rho and promotes *Salmonella* virulence program

Jing Zhang[1,7], Shuo Zhang[1,2,7], Wei Zhou[3,7], Xiang Zhang[4,7], Guanjin Li[1,2], Ruoxuan Li[1,2], Xingyu Lin[1,2], Ziying Chen[1,4], Fang Liu[1], Pan Shen[3], Xiaogen Zhou [5], Yue Gao [3] ✉, Zhenguo Chen [4] ✉, Yanjie Chao [1,2,6] ✉ & Chengyuan Wang [1,2] ✉

Transcription is crucial for the expression of genetic information and its efficient and accurate termination is required for all living organisms. Rho-dependent termination could rapidly terminate unwanted premature RNAs and play important roles in bacterial adaptation to changing environments. Although Rho has been discovered for about five decades, the regulation mechanisms of Rho-dependent termination are still not fully elucidated. Here we report that Rof is a conserved antiterminator and determine the cryogenic electron microscopy structure of Rho-Rof antitermination complex. Rof binds to the open-ring Rho hexamer and inhibits the initiation of Rho-dependent termination. Rof's N-terminal α-helix undergoes conformational changes upon binding with Rho, and is key in facilitating Rof-Rho interactions. Rof binds to Rho's primary binding site (PBS) and excludes Rho from binding with PBS ligand RNA at the initiation step. Further in vivo analyses in *Salmonella* Typhimurium show that Rof is required for virulence gene expression and host cell invasion, unveiling a physiological function of Rof and transcription termination in bacterial pathogenesis.

The Rho factor is a ring-shaped ATP-dependent hexameric helicase that is conserved throughout the bacterial kingdom[1–17]. Rho plays an indispensable function in transcriptional termination and gene regulation in bacteria[1–5]. Coordinated with transcription-translation coupling and mediated by the transcription elongation factor NusG, Rho modulates factor-dependent transcription termination and regulates the gene expression[18–23]. Rho also separates transcription units, represses xenogenic genes[24], silences antisense RNAs[25], removes stalled RNAP from DNA thus maintaining genome stability[26], and prevents

the formation of R-loops[27]. In the model bacterium *Escherichia coli*, about half of the transcription events are terminated by Rho factors[25,26].

Five decades of genetic and biochemical experiments indicate that Rho-dependent transcription termination involves a series of transient steps[1–17]. The Rho-dependent pre-termination complexes were only recently determined at atomic resolution using cryoEM, unveiling the molecular details how Rho initiates transcription termination[20–22]. As the crucial first step, Rho recognizes a long C-rich

[1]CAS Key Laboratory of Molecular Virology and Immunology, Center for Microbes, Development and Health (CMDH), Shanghai Institute of Immunity and Infection, Chinese Academy of Sciences, Shanghai, China. [2]University of Chinese Academy of Sciences, Beijing, China. [3]Department of Pharmaceutical Sciences, Beijing Institute of Radiation Medicine, Beijing, China. [4]The Fifth People's Hospital, Institutes of Biomedical Sciences, School of Public Health, School of Basic Medical Sciences, Fudan University, Shanghai, China. [5]College of Information Engineering, Zhejiang University of Technology, Hangzhou, China. [6]Key Laboratory of RNA Science and Engineering, Shanghai Institute of Biochemistry and Cell Biology, Center for Excellence in Molecular Cell Science, Chinese Academy of Sciences, Shanghai, China. [7]These authors contributed equally: Jing Zhang, Shuo Zhang, Wei Zhou, Xiang Zhang. ✉e-mail: gaoyue@bmi.ac.cn; zhenguochen@fudan.edu.cn; yjchao@ips.ac.cn; cywang@ips.ac.cn

RNA sequence (Rho utilization site; *rut* site) through its PBS and binds with the *rut* site[1–4]. Upon binding, Rho undergoes conformational changes from open-ring state to a catalytically competent, close-ring state. In the final step, Rho performs ATP-hydrolysis-dependent 5′ -> 3′ translocation on mRNA, applying mechanical force to the transcription elongation complex (TEC) and triggering termination. Whereas the latter steps are facilitated by a number of structural and regulatory proteins such as NusG and NusA, whether and how the initial RNA binding step is regulated remains little understood.

Rof (Rho-off, also called YaeO) is the only *E. coli* host factor that directly interacts with Rho and inhibits the Rho-dependent transcription termination[28–30]. Rof was proposed to bind to Rho and inhibit the PBS ligand binding[29]. However, whether Rof forms a protein complex with Rho and the basic nature of the interactions have been elusive. The regulatory and physiological functions of Rof as a conserved host factor in bacteria are unknown.

Herein, we report the atomic cryo-EM structure of Rho-Rof anti-termination complex and reveal the molecular interactions between Rho and Rof. Together with our recently determined Rho-dependent pre-termination complexes[21], the structure shows that Rof directly binds with Rho N-terminal domain (PBS site) and disrupts the interactions between PBS ligand RNA and Rho. Rof regulates the initiation of Rho-dependent termination and inhibits termination efficiency. Our in vivo assays further show that Rof plays crucial functions in virulence regulation in the model bacterial pathogen *Salmonella enterica* serovar Typhimurium. Deletion of *rof* significantly reduce the expression of multiple virulence factors that are required for *Salmonella* invasion of host cells.

## Results

### Structure of Rho-Rof antitermination complex

The Rof and Rho proteins were individually expressed in *E. coli* and purified separately. Gel filtration assays showed that Rof is monomeric while Rho is hexameric in solution (Supplementary Fig. 1a, b). We assembled the Rho-Rof antitermination complex by mixing the purified Rho and Rof by the ratio as 1:1. The assembled complex was further validated in gel filtration assay, and the results confirmed the extensive interactions between Rof and Rho (Supplementary Fig. 1c). We subjected the assembled complex to single-particle reconstruction cryo-EM study and determined a 2.8 Å resolution structure of the Rho-Rof complex (Fig. 1a, Supplementary Fig. 2 and Supplementary Table 1).

The overall structure shows a Rho hexamer (open-ring state, comprising six protomers A-F) interacting with six Rof proteins in each of its protomers (Fig. 1a). The architecture of Rho is identical to previously reported open-ring state Rho structures, in which the midpoints of protomers A and F at either end of the ring are offset by 45 Å and leaves a 12 Å wide gap[15]. High-resolution data clearly show additional Rof density near the N-terminal of Rho with local resolution at ~4–5 Å (Fig. 1a and Supplementary Fig. 2). The densities of Rof in all protomers are clear (we will use the Rof near Rho protomer C and D as an example for the rest part of this paper) except the one near protomer F, enabling unambiguous rigid-body docking of atomic structure of *Ec*Rof (PDB ID: 1SG5). The final models of Rof in the complex exhibit a conserved α-β-sandwich-like structure which was observed in both structures of *Ec*Rof and *Vc*Rof (PDB ID: 6JIE).

Rof shows conformational changes when interacting with Rho. The conformations of Rof structures in our model, particularly the orientation of the N-terminal α-helix (residues 10-21) differ from the NMR structure of *Ec*Rof (PDB ID: 1SG5)[29] and are more similar with the crystal structure of *Vc*Rof (RMSD 3.35 Å vs 3.78 Å, Supplementary Fig. 3a, b). The N-terminal α-helix of Rof in our structure displays a ~ 45° anticlockwise rotation and exhibits a large RMSD value (Supplementary Fig. 3a) when compared to *Ec*Rof (NMR). While the orientation of the N-terminal α-helix is quite similar to that of *Vc*Rof (Supplementary

Fig. 3b). These conformational changes are more likely caused by intermolecular interactions. As in the crystal structure of *Vc*Rof, two *Vc*Rof molecules interact with each other in the asymmetric unit via N-terminal α-helix[30]. In our cryo-EM structure, this α-helix is also involved in the interactions between Rof and Rho (Fig. 1b). In the NMR structure of *Ec*Rof, Rof is monomeric and no protein-protein interactions have been observed[29].

The N-terminal α-helix is crucial in facilitating interactions between Rof and Rho. In our cryo-EM structure, the spatial relationship between each Rof and the Rho protomers is identical. Rof binds extensively to Rho by attaching its N-terminal α-helix to the N-terminal of Rho (Fig. 1b and Supplementary Fig. 3c). The negative-charged residues Asn9, Asp11, Asp14 from Rof N-terminal α-helix form hydrogen bonds and salt bridges with Lys115, Lys105, Ser82, Gln85, Ser84 in Rho, respectively. In addition, residues Cys10 and Tyr13 stabilize the interface via van der Waals interactions with residues Leu114, Phe 89 (Fig. 1c).

The Rof's β3-β4 loops (residues 45–50) make additional interactions with two Rho protomers. As showed in Fig. 1d, the residues Arg46, Lys47, Asn48, and Glu50 from β3-β4 loops make contacts with both protomer D and C by forming strong hydrogen bonds with residues Gly22, Glu19, Glu125, and Arg88, respectively (Fig. 1d).

All residues involved in the Rho-Rof interactions were verified through in vitro binding assays, confirming that Rof's interactions with both protomers are crucial for its binding affinity (Fig. 1e). It appears that the open-ring gap between protomer F and A is causing a disruption in the interactions between Rof's β3-β4 loops and Rho. As a result, the density of Rof in protomer F is considerably worse and has lower local resolution compared to the other protomers of Rho. The residues in the Rho-Rof binding site are conserved across different species, suggesting a general binding mechanism of protein interactions in bacteria (Supplementary Fig. 4).

### Antitermination mechanism of Rof in Rho-dependent termination

The crucial stage of Rho-dependent termination is the binding of the PBS ligand RNA with Rho[2–5]. Previously, we determined structures of Rho-dependent pre-termination complexes[21]. These complexes have NusG bridging Rho and RNAP, have mRNA tracing from RNAP transcription active center and threading through the central channel of Rho, and have 60 nucleotides of RNA interacting sequence-specifically with the exterior of Rho PBS (Fig. 2a, left panel). All six protomers of Rho hexamer interact with the PBS ligand RNA. Each Rho protomer is associated with ten nucleotides, in which 5 nt of the PBS ligand make potentially sequence-specific interactions with Rho protomer and another 5 nt connect the sequence-specifically recognized RNA segment between two protomers. In vitro transcription termination assay shows the PBS ligand is essential for subsequent steps in Rho-dependent termination. Without the PBS ligand, the termination efficiency of Rho-dependent termination is greatly decreased[2–5,21].

Rof competes with the PBS ligand to bind to Rho and inhibits Rho-dependent termination. We superimposed each Rho protomer and corresponding Rof with the Rho-dependent pre-termination complex structure. The Rof binding sites in Rho are in the same location as the PBS ligand RNA according to the superposed model (Fig. 2a, right panel). Zoom-in view reveals the detail geometry of the conflict binding sites. In the Rho-dependent pre-termination complex, the PBS ligand RNA makes sequence-specific interactions with α-helix 4 and β-strands 4-5 of the Rho protomer. In this binding site, 5 nt of PBS-ligand RNA interact with a series of residues in each Rho protomer with a consensus sequence $(AACCC)_6$. The N-terminal α-helix of Rof makes extensive interactions with Rho and almost half of the residues (Ser82, Ser84, Gln85, Arg88, Phe89, and Leu114) are interacting residues involved in the Rho-PBS ligand RNA binding. Molecular modeling suggests that, if α-helix of Rof were present, it would clash with PBS-

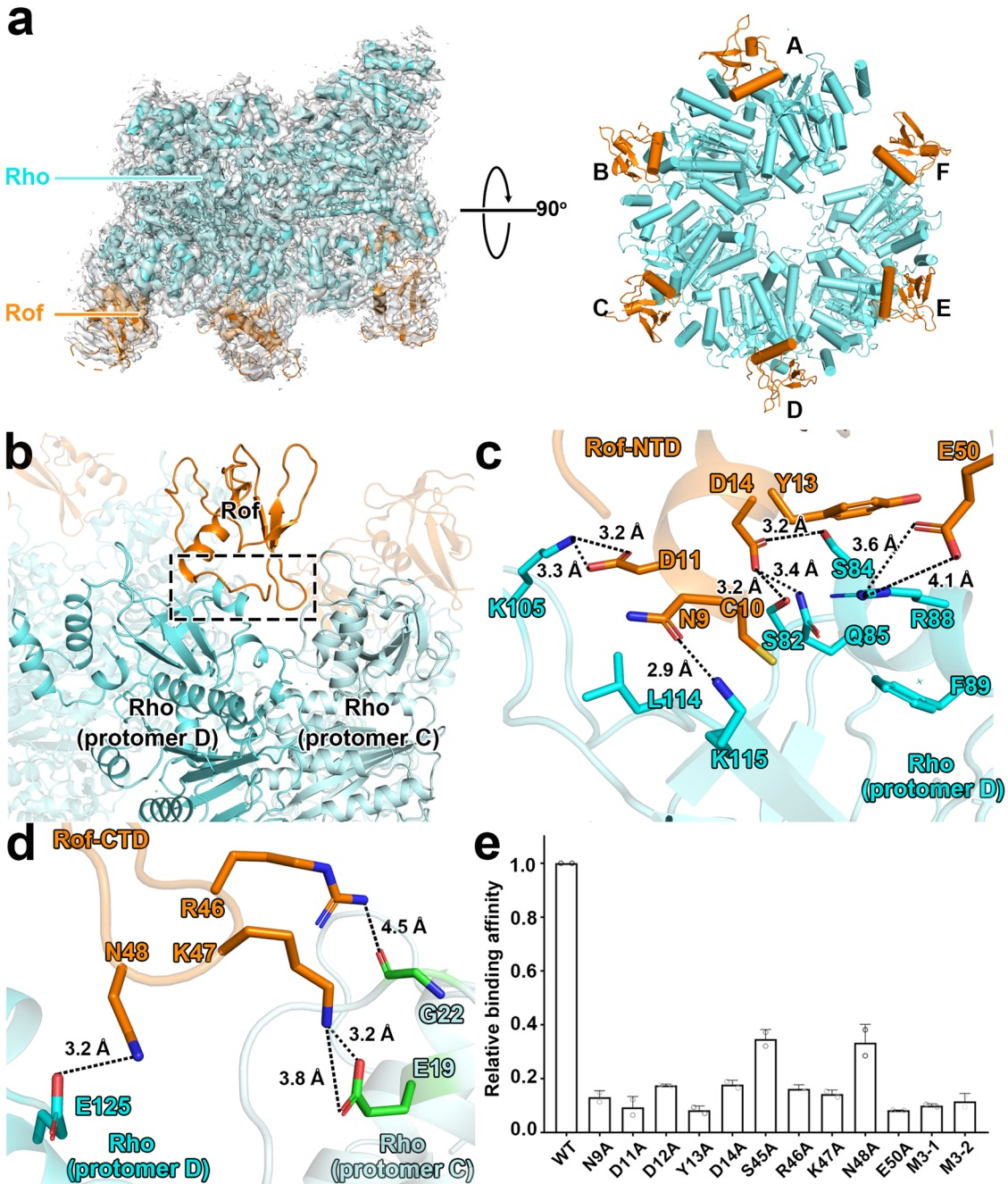

**Fig. 1 | Structure of the Rho-Rof antitermination complex. a** Cryo-EM structure of the Rof-Rho complex. Left panels, view orientation having the Rho hexamer central channel aligned with the y axis, showing the open-ring gap between Rho protomer A and protomer F. Right panels, orthogonal view orientation showing interactions of Rof with Rho protomers A-F. Images show EM density (gray surface) and fit (ribbons) for Rho. Rho and Rof are in cyan and orange. **b** Rho-Rof interactions in Rho-Rof antitermination complex. Overall interactions, focusing on Rof binding with Rho protomer D and C. The interface is highlighted by the dashed black box. Rof, Rho (protomer D) and Rho (protomer C) are in orange, cyan, and light cyan. Other parts of the complex are shown in transparent color. **c** Details of interactions for Rof-NTD and Rho protomer D. Rof residues that interact with Rho are labeled and shown as orange sticks, Rho protomer D residues involving the interactions are shown as cyan sticks. Black dashed lines indicate hydrogen bonds and salt bridges. **d** Details of interactions for Rof-CTD and Rho protomer D and C. Rho protomer C residues involving the interactions are shown as green sticks. Other colors and labels are as Fig. 1c. **e** Substitution of Rof residues involved in interactions with Rho decreased their in vitro binding affinity. Data for Biacore experiments are means of three technical replicates. Error bars represent ± SEM of $n = 2$ experiments. M3-1 represents N9A/D11A/D12A triple mutations, M3-2 represents R46A/K47A/N48A triple mutations.

ligand RNA (Fig. 2b). Electrophoretic Mobility Shift Assay (EMSA) confirms that Rof could compete with PBS-ligand RNA, and disrupt the interactions between PBS ligand and Rho (Fig. 2c, Supplementary Fig. 5a and b). Mutations in Rof's interacting interface could reduce or eliminate binding with Rho and promote PBS ligand binding. In vitro transcription assays confirm that the antitermination efficiency of Rof

is depending on its inhibition efficiency for Rho's binding ability with PBS ligand (Fig. 2c and d).

## Rof is a widely conserved regulator of Rho

As Rho is a highly conserved termination factor, we asked whether Rof is similarly conserved or co-evolved in bacteria. Sequence analyses

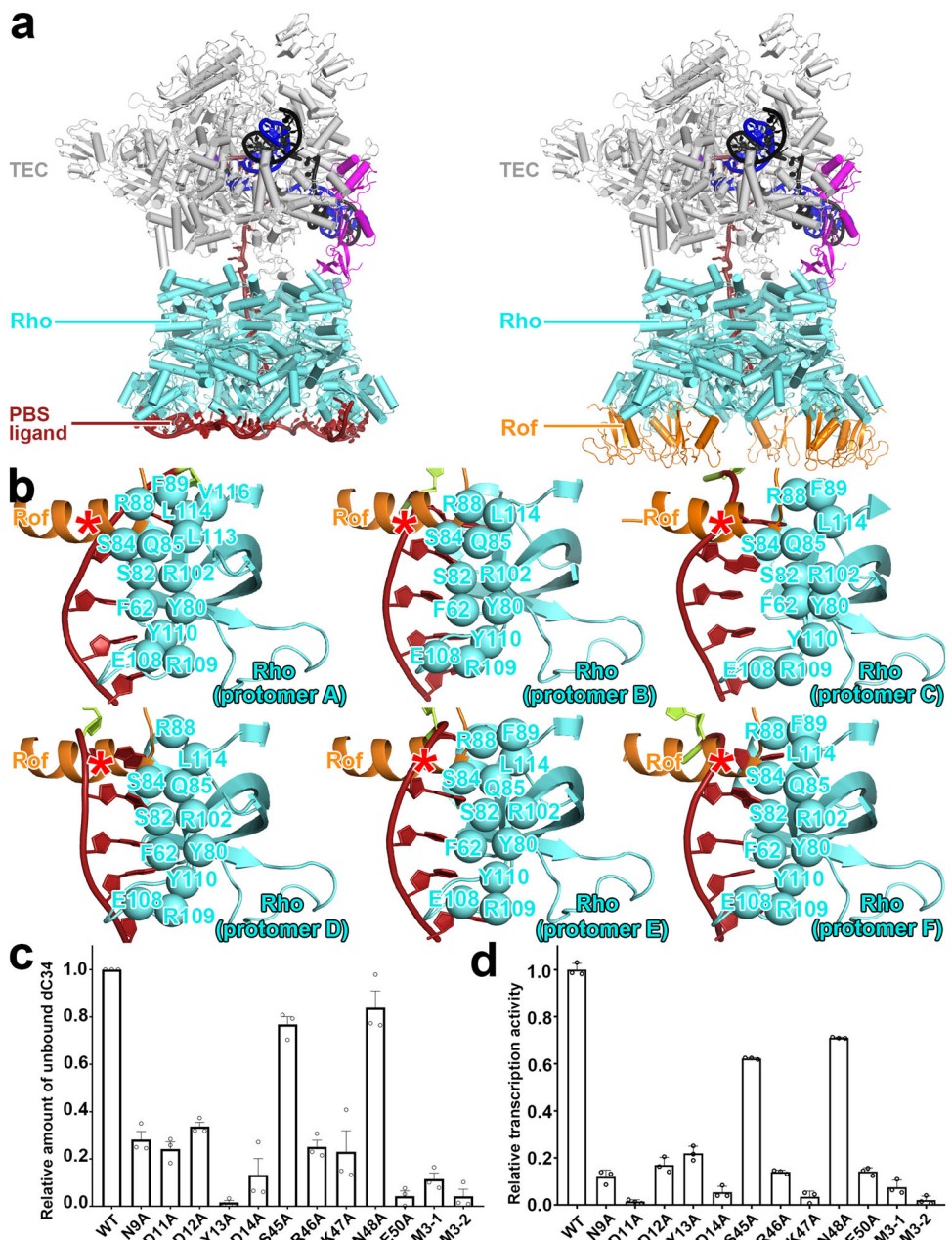

**Fig. 2 | Antitermination mechanism of Rof. a** Superposition of Rho-Rof anti-termination complex with Rho-dependent pre-termination complex. Left panel, overall structures of Rho-dependent pre-termination complex with λtR1 *rut* RNA (λtR1-Rho-NusG-TEC). Rho, cyan; RNAP in TEC, grey; non-template strand DNA, template strand DNA, and RNA, black, blue, and brick-red, respectively; NusG, magenta. Right panel, exactly same view with left panel, each Rof·Rho protomer are superposed with structure of λtR1-Rho-NusG-TEC, and replaced λtR1 *rut* RNA with Rof. Rof, orange; other colors and labels are as left panel. **b** Details of interactions for Rof/λtR1 *rut* RNA with Rho protomers A-F. Rho residues that interact with PBS-ligand RNA are shown as cyan spheres. λtR1 *rut* RNA is shown as brick-red; Rof N-terminal α helix is shown as orange; the competing binding site of Rof and PBS-ligand RNA is marked by red asterisk. **c** Relative inhibition efficiency of Rof on Rho-

dC34 complex identified by electrophoretic mobility shift assay. Substitutions of residues involved in Rof-Rho interactions suppressed Rof's binding affinity with Rho thus decreased Rof's inhibition efficiency on Rho-PBS ligand RNA (dC34) formation. Data for in vitro transcription assays are means of three technical replicates. Error bars represent ± SEM of *n* = 3 experiments. M3-1 represents N9A/D11A/D12A triple mutations, M3-2 represents R46A/K47A/N48A triple mutations. **d** Relative in vitro transcription activity with Rho and Rof. Substitutions of residues involved in Rof-Rho interactions suppressed Rof's antitermination efficiency on Rho-dependent termination thus suppressed in vitro transcription activity. Data for in vitro transcription assays are means of three technical replicates. Error bars represent ± SEM of *n* = 3 experiments. M3-1 represents N9A/D11A/D12A triple mutations, M3-2 represents R46A/K47A/N48A triple mutations.

suggest that *rof* is widely conserved in various bacteria, including Enterobacterales, Neisseriales, and Burkholderiales (Supplementary Fig. 6). Within the enterobacterial family, which contains many bacterial pathogens, the sequence and structure of Rof are highly conserved and show great similarity to the *Ec*Rof (Fig. 3a). For example, the gene sequences of Rho, Rof, and RNAP in *Salmonella enterica* are very

similar to those in *E. coli* (Sequence identity: *rho* 99.5%, *rof* 78.6%, *ropA* 100%, *ropB* 98.7%, *ropC* 98.7%, *ropZ* 100%). It is most likely that Rof in these organisms regulates Rho-dependent termination in a similar manner. Indeed, in vitro transcription assays confirmed that the *Se*Rof could suppress *Se*Rho-mediated transcription termination (Supplementary Fig. 5c). *Se*Rof and *Ec*Rof could act interchangeably in vitro.

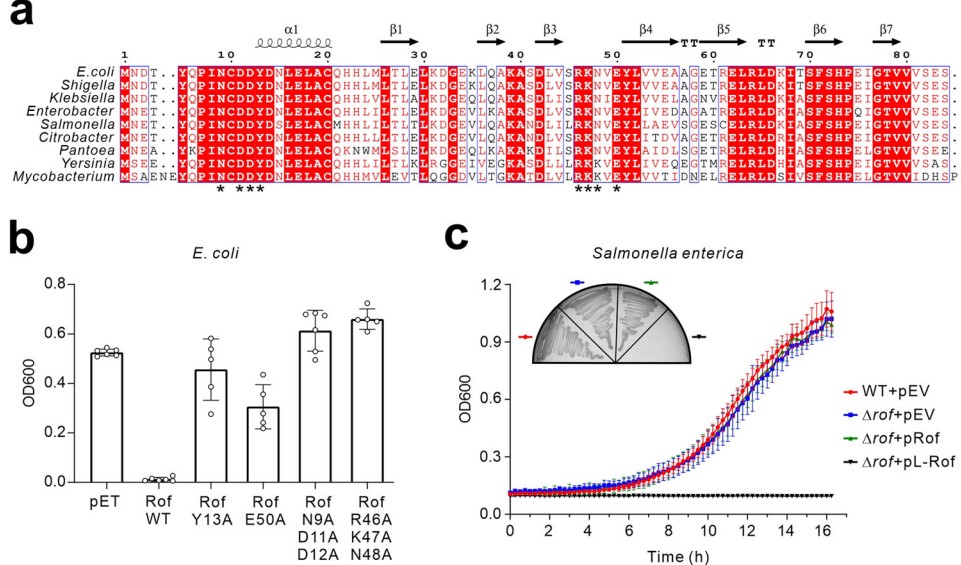

**Fig. 3 | Rof is a conserved regulator of Rho in bacterial pathogens. a** Multiple sequence alignment of Rof. Representative Rof sequences retrieved from UniProt or Genbank are aligned. Secondary structures are shown above, mutated residues used for in vivo testing are marked below by asterisks. **b** Rof inhibits the growth of *E. coli* under osmotic stress. WT and mutant Rof proteins are overexpressed from pET22b vectors, with the empty pET22b plasmid used as vector control (pET). *E. coli* BL21 containing the indicated plasmids are grown overnight in LB supplemented with 0.6 M NaCl and 0.5 mM IPTG at 37 °C. The final OD is recorded and plotted. Data are presented as mean values ± SD from *n* = 6 biology replicates. **c** Rof inhibits

*Salmonella* growth and colony formation in LB. *Salmonella* WT and Δ*rof* mutant are transformed with either empty pZE12 vector (pEV), Rof complementation plasmid (pRof), or Rof overexpression plasmids (pL-Rof). pRof is constructed by cloning the *rof* gene with its own promoter into a promoterless pZE12 vector. pL-Rof contains the *rof* gene driven by a constitutive PLlacO promoter on pZE12. (Figure inset) few or no viable colony on LB plates is obtained after transformation with pL-Rof. After inoculating rare viable colonies into LB liquid medium containing 0.9 M NaCl, bacterial growth is monitored overtime. Data are presented as mean values ± SD from *n* = 3 biology replicates.

We observed that *Se*Rof inhibit *Ec*Rho's termination activity, while *Ec*Rof could also inhibit *Se*Rho's termination activity (Supplementary Fig. 5c). Therefore, Rof proteins utilize a conserved antitermination mechanism to suppress Rho-dependent termination. Interestingly, searching the homologs of Rof and Rho in bacterial genomes revealed that Rof is only present in organisms that encode a conserved copy of *rho* (Supplementary Fig. 7), suggesting that Rof may have emerged later than Rho to regulate termination in response to certain cues or complex environments.

Given the indispensable regulatory function of Rho in bacteria[1–5,31], we next asked whether Rof plays any physiological or regulatory role in bacteria in vivo. Intriguingly, our analysis revealed a regulatory role of Rof in stress responses in *E. coli* and *Salmonella enterica*, two similar and well-established model organisms to study bacterial physiology and pathogenesis, respectively. Whereas an overexpression of *rof* in *E. coli* did not show any detectable phenotype in standard broth, it led to a strong growth inhibition under osmotic stress in LB with 0.6 M NaCl (Fig. 3b). Likewise, *Salmonella* showed an even greater sensitivity to high levels of Rof. A constitutive overexpression of Rof in *Salmonella* inhibited colony formation on LB-agar plates and completely suppressed growth in LB broth (Fig. 3c), while the deletion of *rof* showed little effect. The growth inhibition was relieved in *E. coli* by introducing mutations in conserved key residues in Rof (Fig. 3a, b), supporting that the regulatory role of Rof in stress response is dependent on its binding with Rho.

## Rof regulates the virulence gene expression in *Salmonella* Typhimurium

To further investigate the physiological relevance of Rof at its endogenous levels, we sought to analyze the *Salmonella* Δ*rof* mutant for the expression of virulence genes encoded on the *Salmonella* pathogenicity island-1 (SPI-1), which are specifically induced under osmotic stress with low oxygen[32,33] (similar as the gastrointestinal environment).

Notably, the expression of SPI-1 genes was recently shown to be activated once Rho was genetically inhibited in *Salmonella*[34], supporting our hypothesis that Rof may regulate the SPI-1 virulence program via inhibiting Rho. To address this, we introduced chromosomal 3xFLAGs to several representative genes in the SPI-1 regulatory cascade and examined their protein levels by Western blotting. Three genes were tagged, including *hilA* as the upstream master transcriptional regulator of SPI-1 genes, *prgH* as a structural gene encoding part of the type 3 secretion system (T3SS), and *sopB* encoding an effector protein secreted via the T3SS[33,35,36]. When growing under the SPI-1 inducing condition (LB with 0.3 M NaCl + low oxygen[37]), the Δ*rof* mutant indeed expressed lower amounts of HilA, PrgH and SopB proteins (Fig. 4a–c). The reduced levels in Δ*rof* were fully complemented by a copy of *rof* gene provided in trans (pRof), confirming that Rof positively regulates the SPI-1 virulence gene expression in *Salmonella*. The secretion of SopB-3xFLAG and another untagged effector SipC, were reduced in the Δ*rof* mutant (Fig. 4d, e). The regulation of SPI-1 virulence program is dependent on the Rof-Rho complex formation, since the *rof* triple mutation disrupting the Rho-binding site failed to complement the Δ*rof* mutant (Fig. 4f). The SPI-1 virulence program is required for *Salmonella* to invade eukaryotic host cells[32,35,38]. Using a human colon epithelial cell line HCT116, we have demonstrated that the Δ*rof* mutant has a reduced ability to invade host cells (Fig. 4g). This reduction was rescued by the WT *rof* gene on plasmid but not by the *rof* triple mutant, bolstering a regulatory function of Rof in virulence control in bacterial pathogenesis.

## Discussion

Our work shows the structure of antitermination complex in Rho-dependent termination. The structure elucidates the basic nature of the interactions between Rof and Rho as a bipartite protein complex, in which the N-terminal α-helix of Rof is crucial in facilitating interactions and has conformational changes during the interactions.

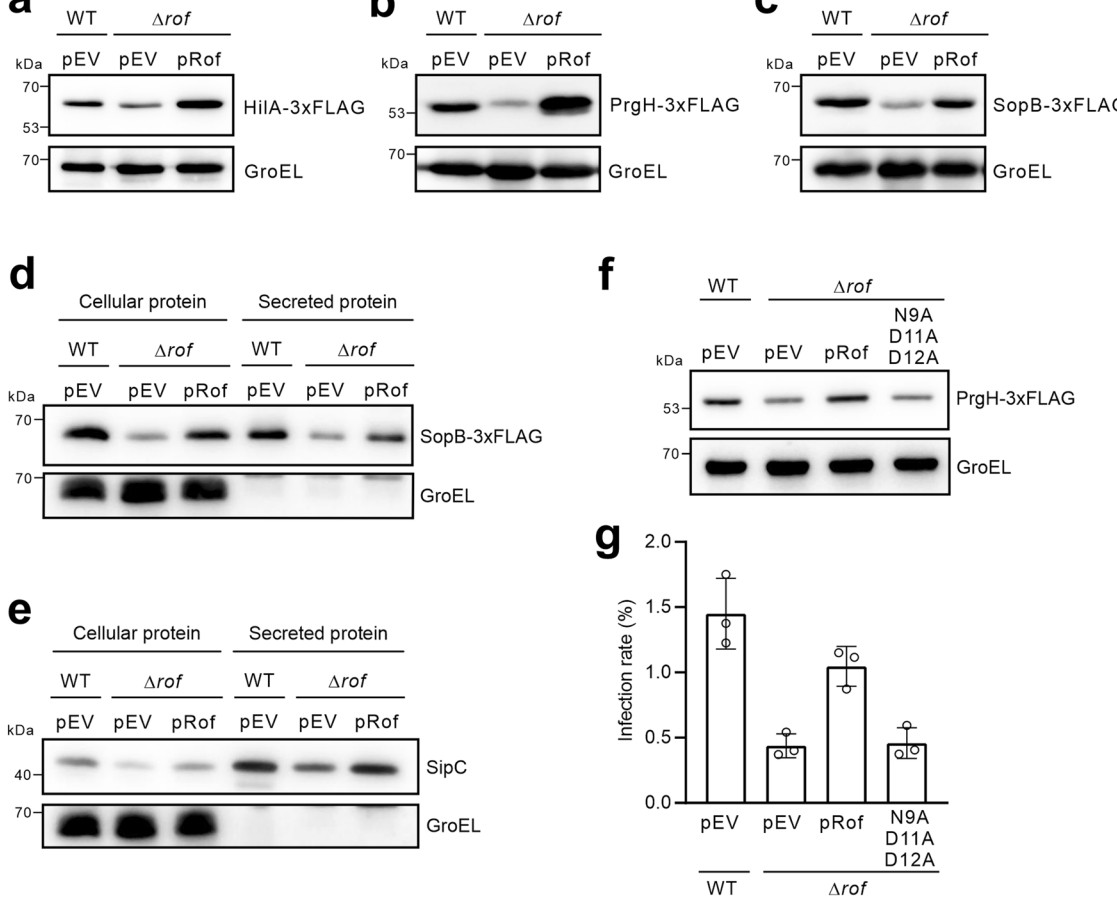

**Fig. 4 | Rof regulates the virulence program of *Salmonella* Typhimurium.** Western blotting analysis of total protein levels or secreted proteins in WT *Salmonella* and Δ*rof* mutants. The secreted proteins are extracted from the culture supernatant. pEV, empty vector. pRof, Rof complementation plasmid, contains a copy of *rof* gene with its own promoter. Total proteins are separated by 12% SDS-PAEG and analyzed using anti-FLAG antibody. GroEL serves as a loading control. Representative data from two independent experiments are shown. Uncropped images are provided as Source Data file. **a**–**c** Western blotting analysis of HilA-3xFLAG protein levels (**a**); PrgH-3xFLAG protein levels (**b**); SopB-3xFLAG protein levels (**c**) in total cellular proteins. **d**–**e** Western blotting analysis of SopB-3xFLAG protein levels (**d**) and SipC protein levels (**e**) in total proteins and secreted proteins. GroEL served as a non-secreted cellular protein control. **f** Western blotting analysis of PrgH-3xFLAG levels in total bacterial proteins. pEV, empty vector. pRof, Rof complementation plasmid, contains a copy of *rof* gene with its own promoter. The indicated point mutations were introduced in pRof. **g** Rof is required for *Salmonella* invasion of epithelial cells. Human HCT-116 cell line was incubated with WT *Salmonella* and *rof* mutants using an MOI of 10. Gentamicin was added to kill extracellular bacteria after 40 min post-infection. The intracellular bacteria were harvested and enumerated after serial dilution and plating on LB agar. Data are presented as mean values ± SD from *n* = 3 biological replicates.

The structure clarifies the antitermination mechanism of Rof, showing that Rof competes with the PBS ligand to bind to Rho. Thus, Rof inhibits the initiation step of Rho-dependent termination. Our in vivo assays further illustrate the regulatory function of Rof in virulence control in bacterial pathogenesis. Based on these results and recent studies, we suggest a model of Rof-Rho in virulence regulation in *S.* Typhimurium. Under OFF conditions, the pervasive transcription of SPI-1 genes (which contains AT-rich sequences) is silenced by both chromatin-binding protein H-NS and Rho[34,39], the latter of which recognizes *rut* site sequence and prevents unwanted transcription of virulence genes (Fig. 5, upper panel). During infection, Rof blocks Rho binding with *rut* site and inhibits Rho-dependent termination, leading to displacement of H-NS from SPI-1 locus, continued transcription of virulence genes, and successful host invasion (Fig. 5, lower panel). These results showed a physiological function of Rof in bacterial pathogens and in virulence regulation.

Though Rof is widely distributed in bacteria that encode the conserved Rho protein, the absence of Rof in certain bacterial lineages indicates that other factors might exist to regulate Rho-dependent termination. Searching homologous structures of Rof using DALI service identified the Sm-like RNA-binding protein Hfq[40,41] on the highest rank. Hfq binds to a large number of small noncoding RNAs and promotes their base-pairing interactions with target mRNAs to facilitate post-transcriptional gene regulation[42,43]. Interestingly, Hfq might interact with Rho, directly or indirectly. Hfq was shown to be co-purified with Rho, and exhibited antitermination function in Rho-dependent termination[44,45]. This suggests that Hfq might be a structural homolog of Rof acting similarly on Rho, despite the low similarity in primary sequence (Sequence identity between *Ec*Rof and *Ec*Hfq, 4.9%, Supplementary Fig. 8). It remains to be clarified whether Rof and Hfq are evolved from the same ancestors or results of convergent evolution, and whether Hfq also inhibits the initiation step of Rho-dependent termination.

Our data suggest that the inhibitory effect of Rof on Rho-dependent termination could be reduced by NusG, which has been shown to bridge Rho and RNAP[2–5]. The interaction of NusG with Rho facilitates the formation of the catalytically competent, closed-ring state of Rho hexamer, triggering transcription termination[4]. By performing in vitro transcription assays with NusG, we observed that the inhibition efficiency of Rof on Rho-dependent termination was

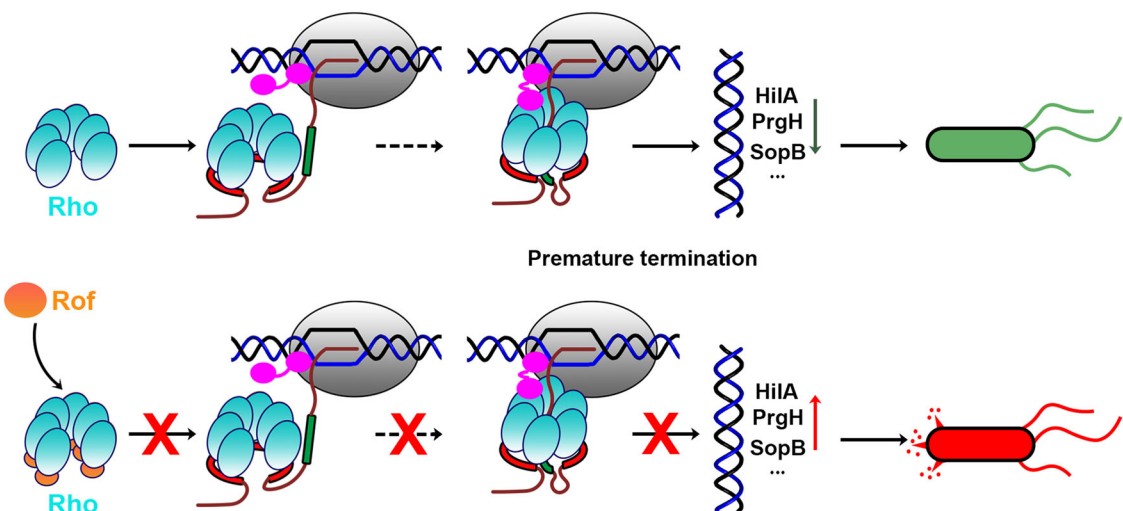

**Fig. 5 | The proposed model of virulence gene regulation in *Salmonella* Typhimurium.** Proposed regulatory mechanism of Rof and Rho in virulence gene expression in *Salmonella* Typhimurium. Rho, cyan; RNAP in TEC, grey; non-template-strand DNA, template-strand DNA, and RNA, black, blue, and brick-red, respectively; PBS ligand, red; SBS ligand, green; RNA 5′ to PBS ligand, RNA between PBS ligand and SBS ligand, and spacer RNA between SBS ligand and TEC, brick-red; NusG, magenta. Upper panel (left to right). In the first step, the Rho hexamer in an open-ring state recognizes *rut* site (PBS ligand) through its primary binding site (PBS) on the exterior of the Rho hexamer. After a series of steps, Rho binds with short pyrimidine-rich RNA sequence (SBS ligand), changes from open-ring state to close-ring state, performs ATP-dependent 5′−3′ translocation on RNA towards the TEC, thus terminates premature transcripts and inhibits spurious transcription of SPI-1 virulence genes such as *hilA, prgH, and sopB* in *Salmonella*. Lower panel (left to right). Rof binds with Rho in its primary binding site (PBS), blocking PBS ligand RNA to bind Rho, thus prevents further transcription termination. The transcription of SPI-1 virulence genes *hilA, prgH, and sopB* are derepressed with reduced Rho-dependent termination.

reduced by the addition of NusG. As shown in Supplementary Fig. 9b, Rof fully inhibit Rho's termination, while NusG could partly restore the termination activity. We also conducted in vitro transcription assays using a triple mutation of Rho (Y80A/R88A/F89A) that eliminates Rho's binding ability with PBS ligand (Supplementary Fig. 9a). The results indicate that Rho[Y80A/R88A/F89A] has reduced termination activity, but NusG can also partially restore it (Supplementary Fig. 9c). Our previous results indicate no direct interaction between Rof and NusG, with Rof binding to Rho's N-terminal domain while NusG binding to Rho's C-terminal domain[21]. It suggests that NusG does not relieve the inhibition of Rof on PBS ligand binding. Interestingly, it has been previously reported that the PBS ligand, ATP, and NusG would trigger Rho ring closure and the ring-closure is the crucial step for Rho-dependent transcription termination as "catch-up" model[46]. Our Rof-Rho complex shows that Rho is in a ring-open state, and following the submission of our manuscript, preprint reports Rof could conformationally regulate the Rho ring dynamics and induce a hybrid conformation[47]. Together with our results that NusG could partially restore the Rho's termination activity which inhibited by Rof, it may suggest the NusG would trigger the Rho ring closure even in the presence of Rof. Further structural analysis of the TEC-Rho-NusG pre-termination complex in the presence of Rof would be needed to make it more clear. Based on these results, we proposed that the initiation of Rho-dependent termination has two distinct mechanisms: one is PBS ligand-dependent initiation which could be regulated by Rof, and the other is PBS ligand-independent initiation which is mediated by NusG. It would be interesting to further dissect these two pathways of Rho-dependent termination and their respective regulatory targets and functions in bacteria in the future.

## Methods
### Protein purification
Rho was purified from BL21(DE3) cells carrying pET24b-*rho* as described previously. 2 L cell culture was grown at 37 °C in LB medium supplemented with kanamycin to $OD_{600} = 0.8$, induced with 1 mM IPTG at 18 °C for 20 h, and harvested. Cells were resuspended in lysis buffer (20 mM Tris-HCl, pH 7.6, 50 mM NaCl, 1 mM EDTA, 1 mM β-mercaptoethanol) and lysed by sonication on ice. Cell lysate was cleared by centrifugation for 30 min at 20,000 g at 4 °C, and applied to a 5 ml HiTrap Q HP column (GE Healthcare) equilibrated with TGE buffer (20 mM Tris-HCl, pH 7.6, 5% (v/v) Glycerol, 1 mM EDTA, 1 mM β-mercaptoethanol) containing 50 mM NaCl. Rho was eluted by a 0.05-0.5 M NaCl gradient over 30 column volumes, concentrated, and diluted with TGE buffer containing 1 mM β-mercaptoethanol to adjust NaCl concentration to 0.05 M. Rho was next applied to a 5 ml HiTrap HP heparin column (GE Healthcare) equilibrated with TGE buffer containing 50 mM NaCl and 1 mM β-mercaptoethanol. Rho was eluted by a 0.05-0.7 M NaCl gradient over 40 column volumes and concentrated to 3 ml volume. Finally, Rho was applied to 120 ml HiLoad 16/60 Superdex 200 size-exclusion column (GE Healthcare) equilibrated with 10 mM Tris-HCl, pH 7.6, 100 mM NaCl, 1 mM β-mercaptoethanol. Fractions containing pure Rho were combined and concentrated using 15 ml Amicon Ultra 100 kDa MWCO concentrators to 20 mg/ml. The product (purity >98%) was stored in aliquots at −80 °C.

Rof was purified from BL21(DE3) cells transformed with pET22b-rof. The protein expression was induced with 1 mM IPTG at 18 °C for 20 h at $OD_{600}$ of 0.7. The cell pellet was lysed in lysis buffer (50 mM Tris-HCl, pH 7.6, 200 mM NaCl, 5% (v/v) glycerol, 0.1 mM PMSF) using sonication. The supernatant was loaded on a 5 mL Ni-NTA column that was subsequently washed and eluted with lysis buffer B containing 200 mM imidazole. The eluted fractions were concentrated to 3 ml and applied to 120 ml HiLoad 16/60 Superdex 200 size-exclusion column (GE Healthcare) equilibrated with 10 mM Tris-HCl, pH 7.6, 100 mM NaCl, 1 mM β-mercaptoethanol. The fractions containing target proteins were concentrated to 4 mg/mL, and stored at −80 °C. Rof derivatives were prepared by the same procedure.

RNAP holo was prepared from BL21 Star (DE3) cells containing pIA900 (encodes *E. coli* RNAP β′ with C-terminal hexahistidine tag, β, α, and ω subunits) and pRSFDuet-rpoD plasmids. The product (purity >95%) was stored in aliquots in RNAP storage buffer (10 mM Tris-HCl, pH 7.6, 100 mM NaCl, 0.1 mM EDTA, and 5 mM dithiothreitol) at −80 °C.

## Cryo-EM sample preparation

For cryo-EM grid preparation, The Rho-Rof complex was freshly prepared as described above and mixed with CHAPSO (Hampton Research, Inc.; final concentration 8 mM) prior to grid preparation. About 3 μL samples were applied to freshly glow-discharged Quantifoil R1.2/1.3 holey carbon grids. After incubation for 1 s at 22 °C and 100% humidity, the grids were blotted for 6 s with blot force 6 in a Thermo Fisher Scientific Vitrobot Mark IV and plunge-frozen in liquid ethane at liquid nitrogen temperature. The grids were prepared in the H2/O2 mixture for 60 s using a Gatan 950 Solarus plasma cleaning system with a power of 5 W. The ø 55/20 mm blotting paper is made by TED PELLA used for plunge freezing.

## Cryo-EM structure determination: data collection and data reduction

Cryo-EM data for Rho-Rof complex were collected at the Fudan University Cryo-EM core Facility, using a 300 kV Titan Krios (FEI/ThermoFisher) electron microscope equipped with a post-GIF Gatan K3 direct electron detector (Gatan). Data were collected automatically in the super-resolution mode, using Serial-EM with a nominal magnification of 105,000x, a calibrated pixel size of 0.595 Å/pixel on the image plane, and with defocus values ranging from −0.8 to −2.0 μm. Each micrograph stack was dose-fractionated to 40 frames with a total electron dose of ~50 e − /Å2 and a total exposure time of 3.6 s. 6033 micrographs of Rho-Rof complex were collected for further processing.

Data were processed as summarized in Figs. S2A-D. Data processing was performed using a Tensor TS4 Linux GPU workstation with four GTX 3090 graphic cards (NVIDIA). Dose weighting motion correction (3×3 tiles; b-factor = 150) were performed using Motioncor2[48]. Contrast-transfer-function (CTF) estimation was performed using CTFFIND-4.1[49]. Subsequent image processing was performed using Relion 3.0[50]. Automatic particle picking with Laplacian-of-Gaussian filtering yielded an initial set of 2,399,274 particles. Particles were extracted into 500 × 500 pixel boxes and subjected to rounds of reference-free 2D classification and removal of poorly populated classes, yielding a selected set of 1,099,028 particles. The selected set was auto-refined using a mask with a diameter of 300 Å, yielding a reconstruction at 3.3 Å overall resolution. The resulting 3D auto-refined particles were further done with post-processing with a soft masked, yielding reconstructions at 2.8 Å.

The initial atomic model for Rho-Rof complex was built by manual docking using open-ring Rho (PDB ID: 1PV4 https://www1.rcsb.org/structure/1PV4) and a crystal structure of E. coli Rof (PDB ID 1SG5 https://www1.rcsb.org/structure/1SG5) in UCSF Chimera[51]. For Rho (residues 280-284) and Rof (residues 1-5), density was absent, suggesting high segmental flexibility; these segments were not fitted. Refinement of the initial model was performed using real_space_refine under Phenix. The Rof and Rho were rigid-body refined against the map, followed by real-space refinement with geometry, rotamer, Ramachandran-plot, Cβ, non-crystallographic-symmetry, secondary-structure, and reference-structure (initial model as reference) restraints, followed by global minimization and local-rotamer fitting. Secondary-structure annotation was inspected and edited using UCSF Chimera. Rof were subjected to iterative cycles of model building and refinement in Coot[52]. The final atomic model at was deposited in the Electron Microscopy Data Bank (EMDB) and the Protein Data Bank (PDB) with accession codes EMDB 37342 and PDB 8W8D (Supplementary Table 1).

## Sequence alignment

Representative Rof and Rho sequences were retrieved from UniProt (https://www.uniprot.org/) or NCBI. All the sequences were aligned by using ClustalX, and draw the final alignment using ESPript 3.0 online tool (https://espript.ibcp.fr/ESPript/ESPript/). The structural information of Escherichia_coli Rof (PDB 1SG5) and Rho (PDB ID: 1PV4) were used to generate the secondary structure in the final alignment.

## Electromobility Gel Shift Assays

Assays of nucleic acid binding to Rho were carried out as described in a previous report[53]. Chemically synthesized oligonucleotide dC34 (3 uM, final concentration, Sangon Biotech) and Rho (hexamer, 3 uM, final concentration) were mixed with Rof or its derivatives (monomer, 36 uM, final concentration) in the reaction buffer (25 mM Tris-HCl, pH 8.0, 5 mM MgCl$_2$, and 50 mM KCl, 10% glycerol). Bovine serum albumin (BSA, 200ug/ml, final concentration) was used as a non-specific protein competitor. Each sample was incubated at 37°C for 15 min and then analysed on a 6% polyacrylamide gel (acrylamide/bisacrylamide ratio of 29:1) in TBE (90 mM Tris, 44 mM boric acid, 2 mM EDTA) followed by SYBR-Gold staining. DNA bands were visualised using a gel documentation system (Tanon 2500 R), and their intensities were quantified using ImageJ software (NIH Image, National Institutes of Health, Bethesda, MD, USA; online at: http://rsbweb.nih.gov/ij/).

## Fluorescence-detected in vitro transcription assay

The experiment was performed as in a previous report[54]. Briefly, E.coli RNAP holo (final concentration: 50 nM) was incubated with DNA template (final concentration: 16 nM) in the reaction buffer at 37°Cfor 15 min. Then, Rho (hexamer, 240 nM, final concentration) with or without Rof (monomer, 8.64 uM, final concentration) was added and incubated at 37°C for 10 min. Finally, TOl-3PEG-Biotin (final concentration: 0.5 uM) was added and the reactions were initiated by the addition of NTP mix (final concentration: 0.4 mM ATP, GTP, and CTP and 0.004 mM UTP). The fluorescence signals were collected using Gen5 (version 3.12) software on a plate reader (BioTek synergy h1) at an excitation wavelength of 510/9.5 nm and an emission wavelength of 550/9.5 nm.

## Biacore

BIAcoreT200 was used to study the direct binding of Rof or its derivatives to Rho. Rho was immobilized (Target: 1000 resonance units (RU), Real: 1176 RU) by amine group coupling on research-grade CM5 sensor chips at pH 4.5. Rof and its derivatives in HEPES buffer (10 mM HEPES (pH 7.4), 300 mM NaCl, and 0.05% (v/v) Tween-20) with a continuous series of concentrations (0, 0.375, 0.75, 1.5, 3, 6, 12, 24, 48 uM) were then injected over the immobilized Rho at 25°C. Kinetics constants (KD) were derived by Scatchard analysis or nonlinear curve fitting of the standard Langmuir binding isotherm using Biacore evaluation Software version 3.2.1 (BIAcoreT200).

## Salmonella strains and growth conditions

Salmonella enterica serovar Typhimurium strain SL1344 was used as wild-type. Strains with deletions or chromosomally 3xFLAG epitope-tagging were constructed using the λ-Red recombinase method. Phage P22 was used to transduce chromosomal modifications to desired genetic backgrounds. Resistance markers were eliminated using pCP20 expressing a FLP recombinase. For plasmid construction, the desired gene fragments were generated by PCR amplification using S. Typhimurium SL1344 or E. coli K-12 MG1655 genomic DNA as template and, after digestion with restriction enzymes, were cloned into the corresponding sites of the indicated vectors. Mutations and plasmid inserts were confirmed by Sanger sequencing.

Salmonella was routinely cultured at 37 °C on LB agar or with shaking at 220 rpm in liquid LB media (10 g/L tryptone, 5 g/L yeast extract, 5 g/L NaCl). Overnight cultures were grown from a single

colony, diluted 1:100 in fresh medium, and grown to the indicated $OD_{600}$. For SPI-1 condition culture, a single colony was inoculated into 15 ml LB containing 0.3 M NaCl in tightly sealed 15 ml Falcon tubes, incubated overnight at 37 °C without agitation. The following compounds were added at the following final concentrations when appropriate: IPTG at 0.5 mM; ampicillin (Amp) at 100 µg/mL; kanamycin (Kan) at 50 µg/ml, chloramphenicol (Cm) at 20 µg/ml; hygromycin (Hyg) at 100 µg/ml.

### Extraction of secretory proteins from culture medium

*Salmonella* was cultured overnight under the SPI-1 condition. 2 ml culture was centrifuged at 12000 rpm for 5 min at room temperature. 1.5 ml supernatant was carefully transferred into a new tube, and added with 0.25 vol of 50 % TCA (Trichloroacetic acid). The mixture was incubated at 4 °C for 1 h, then the protein pellet was collected by centrifugation at 12000 rpm at 4 °C for 30 min. The pellet was washed twice with ice-cold acetone, and air-dried completed until the acetone was evaporated. The protein pellet was finally dissolved in 20 µl 1× protein loading buffer.

### Western blotting analysis

Bacterial samples were resuspended in 1× protein loading buffer and boiled at 98 °C for 8 min. 0.05 OD of proteins samples were loaded each lane. Proteins were transferred onto PVDF membranes (#10600023, Cytiva) for 45 min at 150 V in transfer buffer. Membranes were blocked for 1 h at room temperature in 1× TBST buffer with 5% (w/v) skim milk (#A600669-0250, Sangon), washed with TBST twice and incubated with monoclonal α-FLAG (Sigma-Aldrich #F1804-5MG; 1:10,000), α-SipC (1: 3,000) or α-GroEL (Sigma-Aldrich #G6532; 1:10,000) antibodies for 1 h at room temperature. After three TBST washes, membranes were incubated with secondary α-mouse or α-rabbit HRP-linked antibodies (Sangon #D110087 or #D110058; 1:10,000) for 1 h at room temperature. Chemiluminescence was developed using the high sensitive ECL luminescence reagent (#C500044-0100, Sangon), and then visualized on ChemiScope 6000SE and quantified using ImageJ Software.

### *Salmonella* infection assay

Human colon cancer cell line HCT116 was purchased from CCTCC (#SCSP-644, Shanghai, China). The cells were grown in high-glucose Dulbecco's Modified Eagle's (DMEM) Medium (#12491015, Gibco) supplemented with 10% fetal bovine serum (FBS) (#2534387, Gibco), 50 U/ml penicillin and 50 µg/ml streptomycin at 37 °C with 5% CO2 and humidified atmosphere. Before infection, $1\times10^5$ HCT116 cells were seeded in 1 ml DMEM in 12-well microplates. *Salmonella* was grown under the SPI-1 condition. Bacterial cells were harvested by centrifugation (2 min at 12,000 rpm, room temperature), washed twice with phosphate-buffered saline (PBS, #B548117-0500) and resuspended in DMEM. Infection of HCT116 cells was carried out by adding the bacterial suspension directly to each well, at a multiplicity of infection (MOI) of 10. Cells were then incubated for 40 min at 37 °C in 5% CO2 and humidified atmosphere. After washing twice with PBS, cells were added with fresh DMEM containing 100 µg/ml gentamicin, and incubated for 1 h to kill extracellular bacteria. Afterwards, the infected cells were washed two times with PBS, and added with fresh DMEM containing 50 µg/ml gentamicin for 1 h. The infected cells were washed with PBS and lysed by the addition of PBS with 0.1 % Triton X-100 (#A600198-0500, Sangon). After serial dilution of the lysate in PBS, intracellular bacteria were enumerated on LB agar plates after incubation overnight at 37 °C.

### Reporting summary

Further information on research design is available in the Nature Portfolio Reporting Summary linked to this article.

## Data availability

Cryo-EM map have been deposited in the Electron Microscopy Database (EMDB accession code EMD-37342), and atomic coordinate have been deposited in the Protein Database (PDB accession code 8W8D). Unique biological materials will be made available upon request. Source data are provided with this paper.

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

## Acknowledgements

We thank Dr. Jörg Vogel for the anti-SipC antibody. We thank Dr. Liangliang Kong, Dr. Fangfang Wang and Dr. Jialin Duan at the Electron Microscopy System of the National Facility for Protein Science in Shanghai (NFPS), for the assistance with cryo-EM data collection. We thank Jiang Shao for the experimental support. This study was financially supported by the National Key R&D Program of China (2022YFC2303200 and 2022YFE0111800 to Y.C.), the Natural Science Foundation of China (32270039 to C.W., 32270064 to Y.C., 62203389 to X.Z., 31970146 to Z.C.), Shanghai Pujiang Program (21PJ1414700 to C.W.), Shanghai Science and Technology Innovation Action Plan 2023 "Basic Research Project" (23JC1404201 to C.W.), Shanghai Municipal Science and Technology Commission (2019SHZDZX02 to C.W. and Y.C.), Chinese Academy of Sciences (176002GJHZ2022022MI to Y.C.). Innovation Team and Talents Cultivation Program of the National Administration of Traditional Chinese Medicine (ZYYCXTD-D-202207 to Y.G.), the Young Elite Scientists Sponsorship Program by CAST (2021-QNRC1-03 to W.Z.).

## Author contributions

J.Z., Y.G., Y.C., C.W. designed experiments. J.Z., W.Z., G.L., R.L., P.S. prepared proteins and performed biochemical experiments. J.Z., X.Z. performed cryo-EM data collection. S.Z., X.L., Z.C, F.L. constructed bacterial mutants and performed gene expression and infection experiments. X.G.Z. Y.G., Z.G.C., Y.C., C.W. analyzed data. Y.C. supervised in vivo functional analyses in *E. coli* and *S. enterica*. C.W. supervised biochemical and structural analyses in vitro. Y.C., and C.W. prepared figures and wrote the manuscript.

## Competing interests

The authors declare no competing interests.
