## [Peer Review File · Nature Communications]

A widely conserved protein Rof inhibits transcription termination factor Rho and promotes Salmonella virulence programREVIEWER COMMENTS

Reviewer #1 (Remarks to the Author):

Zhang ¹¹¹_{SEP} et al examine the mechanism of action of the E coli Rof protein, an inhibitor the Rho transcription terminator, by cro-EM, biochemical and in vivo assays. The various approaches are consistent: Rof prevents closure of the Rho hexamer ring. The work is thorough and convincing but needs clarification of the points listed below

1. Are Hfq and Rof redundant? Is a hfq null mutant synthetic lethal with a rof null?
2. It is suggested that NusG promotes Rho-dependent termination via a Rho PBS-independent pathway. The authors need to show this directly by measuring termination in vitro in Rho PBS mutants +/- NusG .
3. Fig. 3 The Salmonella results are puzzling. A rof null and Rof OE appear to have the same phenotype.
4. “The Δ rof mutant indeed expressed lower amounts of HilA, PrgH and SopB proteins (Fig. 3 c-e)”
“The virulence genes hilA, prgH, and sopB are expressed without the inhibition of Rho-dependent termination.”

Taken together, these two conclusions are contradictory.

The authors seem to be saying that the expression of certain genes requires Rho-dependent transcription termination, a counter-intuitive conclusion to say the least.

Are the virulence genes rich in rare codons and wont be expressed if Rho is not inhibited (analogous to the role of RfhA in E coli which specifically blocks Rho in related operons).

Reviewer #2 (Remarks to the Author):

The Authors here report the first cryo-EM structure of ρ /Rof complex describing the antitermination mechanism driven by Rof, which has not been well characterized before. ρ is an essential global gene regulator in most bacterial species that performs its regulation by silencing useless or harmful RNA

synthesis. Rof antitermination factor acts as a PBS ligand blocking PBS/RNA interactions, preventing ρ from terminating the transcription. In the study, they describe the structural details of ρ /Rof interactions and propose the Rof-dependent mechanism of antitermination. Also, the authors claim that Rof-dependent antitermination is happening due to the blocking of PBS by Rof and preventing RNA binding to the same sites. ρ /Rof interactions and involved residues are further confirmed by mutational studies of all involved amino acid residues and affinity testing of Rof binding. The proposed mechanism is supported by a comparison of two complexes and an obvious clash between ρ residues interacting with Rof or RNA. Also, the authors performed EMSA analysis to confirm that Rof blocks PBS sites and prevents RNA binding together with in vitro transcription to test if such blocking affects the proposed mechanism. Additionally, the authors studied Rof role in Salmonella Typhimurium and unraveled that Rof controls virulence gene expression in Salmonella, and furthermore, overexpression of Rof protein completely inhibits Salmonella growth and colony formation. In general, this study well described the structural details of ρ /Rof and proposed the Rof-dependent mechanisms of antitermination, which were supported by biochemical and genetic functional evidence.

There are several concerns that need to be addressed. Notably, this reviewer cannot see the relevant/causal/logical connection between the structure of E. coli Rho-Rof and the mechanism of the transcription of Salmonella Typhimurium. All the conclusions, presumptions, and mechanisms in Salmonella Typhimurium, which are related to/based on E. coli Rho-Rof, are questionable and need additional data support (see below). Specific concerns are outlined below:

Major Concerns:

1. Rof and Rho are from E. coli, However, the authors provided in vivo data from Salmonella Typhimurium and proposed many mechanisms about the transcription of Salmonella Typhimurium. In Figure S4, the authors showed Rho and Rof are highly conserved across strains and the Rho-Rof interface residues are also highly conserved. However, it is not clear if the structures and sequences of E. coli RNAP are highly similar to those of Salmonella Typhimurium RNAP. In addition, we don't know if Salmonella Typhimurium Rho and Rof would form the same complex as that of E. coli. Thus, all the functional conclusions and mechanisms about the transcription of Salmonella Typhimurium are unpersuadable. Authors need to provide additional relevant evidence to connect their presumptions on the transcription of Salmonella Typhimurium with the determined E. coli Rof-Rho complex.

2. "Rof's N-terminal α -helix undergoes conformational changes upon binding with Rho" Figure S3 shows the superimposition between the Apo-Rof and Rof in this complex, however, it is hard to recognize what the conformational change they refer to. In this reviewer's opinion, they share similar conformation based on the provided figures. A clearer and more convincing comparison needs to be provided.

3. About cryo-EM data: (The overall cryo-EM map quality is high; the density of the interface is clear in the map.)

(1) Figures of model-to-map FSCs need to be present to demonstrate the quality of the models and the corresponding ones (model resolution when FSC=0.5) should also be included in Table 1. The accuracy of the model is important for structural explanations and description.

(2) In addition, supplementary figures of local density maps with different regions of the models, showing sidechains, are necessary and important to justify the claimed 2.8Å resolution.

(3) Figures or panels showing high-quality density of interface residues' side chains should be provided to support the structural observations. Authors could add the map to figure 1 or present it in a supplementary figure.

(4) Why has the evaluation information of cryo-EM map been deleted from the official validation report?

(5) Table S1 has missed many necessary items, which need to be provided.

(6) It seems that cryo-EM data was processed using RELION. However, it's puzzling that the authors did not mention and cite RELION.

Minors:

1. In the abstract: "The structure shows that Rof binds to the open-ring Rho hexamer and inhibits the initiation of Rho-dependent termination" – please rewrite it. The structure doesn't directly show Rof inhibits termination.

2. Please also provide PDB code for VcRof.

3. It is better to mention in the very beginning of the detailed description of the ρ /Rof interaction interface that you are going to use a particular example between ρ protomers C and D but not at the end of the description.

4. Figure adjustments:

(1) Potential (my opinion). Figure 1a: The resolution of the cryo-EM map used for this figure seems significantly lower than that provided for the reviewers. Please update the figure with a better representation.

(2) Figure 1d needs a protomer D label.

(3) Supplementary Figure 1 compares the assembly of ρ /Rof and individual components that may benefit from the addition of a chromatogram with a molecular weight standard.

(4) Supplementary Figure 3A: The choice of colors for two superimposed structures is not clearly distinguishable please choose more contrast (my personal opinion) (yellow and blue for example).

5. Page 5 line 126. typo Ser8e

Reviewer #3 (Remarks to the Author):

In the manuscript titled 'Rof is a key regulator of Rho and transcription termination in bacteria', Zhang, Zhang, Zhou, and Zhang et al, showed the cryo-EM structure of the Rho-Rof complex and suggested that the Rof inhibits Rho-dependent termination by binding to the PBS (primary RNA binding site) of Rho. Then, the authors switch gears and provide the first biochemical evidence that the Rof regulates the gene expression of virulence factors of Salmonella. Although the individual experiment is valid and provides useful insights into the function of Rof, this reviewer wonders why the authors suddenly shifted the goal of the manuscript from describing the molecular mechanism of anti-termination by Rof using high-resolution structure of E. coli Rho-Rof complex to validating the relation between Rof and virulence factor expression in Salmonella. Addressing both themes is meaningful, but the breadth of research and the depth of discussion on each topic are somewhat lacking.

Major points

1. The title is too broad, providing little information about the manuscript. This reviewer suggests changing the title containing the key idea of the manuscript.
2. On page 5 and Fig 2d, the in vitro transcription assay result is reported, but there is no gel image. This reviewer suggests putting the transcription assay gel image in the supplementary data. This applied to Fig. S10 as well. This is critical because the transcription assay results can be analyzed in any way, and you never know the result correctly until you see the gel itself. Particularly, previous literature (<https://onlinelibrary.wiley.com/doi/epdf/10.1046/j.1365-2958.1998.00981.x>) reported that the Rof function could not be confirmed in vitro.
3. In the Fig 3b (and Fig S6a), the colony formation of Salmonella Typhimurium was normal in Δ rof + pRof but lost in Δ rof + pL-Rof. This reviewer could not find what pRof is. In addition, main text writes "overexpression of Rof inhibited E. coli growth under osmotic stress (Fig. 3a) and completely inhibited Salmonella growth and colony formation in standard LB medium (Fig. 3b, Supplementary Fig 6).", but the graph in Fig3b seems to suggest high salt does not affect the Salmonella cell growth both in WT and Δ rof. In addition, the figure legend does not comment on this graph.
4. The authors claim that the results show that the Rof regulates virulence gene expression, in particular, SPI-1 genes for the first time. To solidify the function of the Rof in pathogenesis, this reviewer suggests

the authors comment on the upstream pathway – for example, how the Rof is turned on (transcriptional regulation?).

5. This reviewer believes that the first part of the manuscript, concerning the Rho-Rof complex structure, could benefit from a more exploration of different perspectives. For example, did Rof induces any conformational change in Rho? If so, how would that affect the termination?

Minor points

1. Line numbers in the manuscript are necessary for referencing specific content within the text.

2. On page 3-4, “High-resolution data clearly show additional Rof density near the N-terminal of Rho with local resolution at ~4-5 Å (Fig. 1a and Supplementary Fig. 2). The densities of Rof in all protomers are clear except the one near protomer F, enabling unambiguous rigid-body docking of atomic structure of EcRof (PDB ID: 1SG5).”. In this reviewer’s observation, the densities of the two (not only one) Rof protomers on the Rho protomers beside the opening are less clear than the other four, and as the authors commented one is much worse than the other. Do you think this is due to the occupancy or flexibility of the molecules? Why?

3. On page 7, “This suggests that Hfq might be a structural homolog of Rof acting similarly on Rho, despite the low similarity in primary sequence (Supplementary Fig. 9).”. However, in Fig S9, the sequence homology looks pretty high. Could you provide the exact percentage of the sequence homology?

4. In discussion, “Based on these results, we proposed that the initiation of Rho-dependent termination has two distinct mechanisms: one is PBS ligand-dependent initiation which could be regulated by Rof, and the other is PBS ligand-independent initiation which is mediated by NusG.” Do the authors think these relate to ‘catch-up (or RNA-dependent) termination’ and ‘stand-by (or RNAP-dependent) termination’ (<https://doi.org/10.1038/s41467-022-29321-5>)?

5. The Rof forms a stable complex with rho hexamer regardless of the RNA ligand. Then, how would Rof differentiate the virulence-related genes from other housekeeping genes? If this manuscript is the first report showing that the Rof regulates virulence-related gene expression, this might need to be suggested at least in the Discussion.

6. The structure of E.coli Rof is solved, but only Salmonella Rof function is described. Could you (at least briefly) suggest the physiological role of Rof in E.coli?

7. Page 3, previous reported  previously reported?

8. In Fig. S3, How about drawing a figure showing the RMSD values of Ca in Rof. Either the color or thickness of the putty can describe the large RMSD values (<https://pymolwiki.org/index.php/ColorByRMSD> or https://pymolwiki.org/index.php/File:B_factor_putty.png). If one Rof in the protomer F deviates from others, draw the one and compare it with the other five might show it more clearly.

RESPONSES TO REVIEWERS

Reviewer #1 (Remarks to the Author):

Zhang et al examine the mechanism of action of the E coli Rof protein, an inhibitor the Rho transcription terminator, by cro-EM, biochemical and in vivo assays. The various approaches are consistent: Rof prevents closure of the Rho hexamer ring. The work is thorough and convincing but needs clarification of the points listed below

We thank the reviewer for the favorable assessment.

1. Are Hfq and Rof redundant? Is a hfq null mutant synthetic lethal with a rof null?

Thanks for this interesting question. To address it, we successfully constructed a $\Delta hfq \Delta rof$ double mutant in *Salmonella*. As shown in the figure below, the double mutant is viable and shows no obvious growth defect compared to the parental Δhfq single mutant. Therefore, the *hfq* null mutant is not synthetic lethal with the *rof* null mutant.

2. It is suggested that NusG promotes Rho-dependent termination via a Rho PBS-independent pathway. The authors need to show this directly by measuring termination in vitro in Rho PBS mutants +/- NusG .

Thanks for the suggestion. We address this point in the current version (line 226-229): “We also conducted in vitro transcription assays using a triple mutation of Rho (Y80A/R88A/F89A) that eliminates Rho's binding ability with PBS ligand (Supplementary Fig. 9a). The results indicate that RhoY80A/R88A/F89A has reduced termination activity, but NusG can also partially restore it (Supplementary Fig. 9c). ”

3. Fig. 3 The *Salmonella* results are puzzling. A *rof* null and Rof OE appear to have the same phenotype.

We apologize for the confusion. Actually, the *rof* null and the Rof OE mutants have distinct and opposite phenotypes. For example, the *rof* null mutant shows no growth defect in *Salmonella* under stress conditions, but the Rof OE mutants (strain with the pL-rof plasmid) inhibit the bacterial growth (Figure 3c). As another example, the *Salmonella rof* null mutant has reduced expression of SPI-1 virulence genes, whereas the SPI-1 expression was restored to the WT levels by Rof complementation (with pRof rescue plasmids, in which the

constitutive promoter on the pZE12 plasmid was replaced with the own promoter of *rof*, so that *Rof* would not be overexpressed to avoid its lethal effect on bacteria).

4. “The Δ *rof* mutant indeed expressed lower amounts of *HilA*, *PrgH* and *SopB* proteins (Fig. 3 c-e)”

“ The virulence genes *hilA*, *prgH*, and *sopB* are expressed without the inhibition of Rho-dependent termination.”

Taken together, these two conclusions are contradictory.

The authors seem to be saying that the expression of certain genes requires Rho-dependent transcription termination, a counter-intuitive conclusion to say the least.

We apologize for this contradictory argument regarding our model in the figure legend. We have re-written this sentence in the legend of Figure 5: “The transcription of SPI-1 virulence genes *hilA*, *prgH*, and *sopB* are derepressed with reduced Rho-dependent termination”.

Are the virulence genes rich in rare codons and wont be expressed if Rho is not inhibited (analogous to the role of *RfhA* in *E coli* which specifically blocks Rho in related operons).

The SPI-1 virulence genes are horizontally acquired with AT-rich in sequences. A recent study published in PNAS discovered that their spurious expression was silenced by both H-NS and Rho together. We have referred to this model of mechanism in the revised manuscript (line 202-203):“ Under OFF conditions, the pervasive transcription of SPI-1 genes (which contains AT-rich sequences) is silenced by both chromatin-binding protein H-NS and Rho^{34,39}.”

Reviewer #2 (Remarks to the Author):

The Authors here report the first cryo-EM structure of ρ /*Rof* complex describing the antitermination mechanism driven by *Rof*, which has not been well characterized before. ρ is an essential global gene regulator in most bacterial species that performs its regulation by silencing useless or harmful RNA synthesis. *Rof* antitermination factor acts as a PBS ligand blocking PBS/RNA interactions, preventing ρ from terminating the transcription. In the study, they describe the structural details of ρ /*Rof* interactions and propose the *Rof*-dependent mechanism of antitermination. Also, the authors claim that *Rof*-dependent antitermination is happening due to the blocking of PBS by *Rof* and preventing RNA binding to the same sites. ρ /*Rof* interactions and involved residues are further confirmed by mutational studies of all involved amino acid residues and affinity testing of *Rof* binding. The proposed mechanism is supported by a comparison of two complexes and an obvious clash between ρ residues interacting with *Rof* or RNA. Also, the authors performed EMSA analysis to confirm that *Rof* blocks PBS sites and prevents RNA binding together with in vitro transcription to test if such blocking affects the proposed mechanism. Additionally, the authors studied *Rof* role in *Salmonella Typhimurium* and unraveled that *Rof* controls virulence gene expression in *Salmonella*, and furthermore, overexpression of *Rof* protein completely inhibits *Salmonella* growth and colony formation. In general, this study well described the structural details of ρ /*Rof* and proposed the *Rof*-dependent mechanisms of antitermination, which were supported by biochemical and genetic functional evidence.

We thank the reviewer for the favorable assessment.

There are several concerns that need to be addressed. Notably, this reviewer cannot see the relevant/causal/logical connection between the structure of *E. coli* Rho-Rof and the mechanism of the transcription of *Salmonella* Typhimurium. All the conclusions, presumptions, and mechanisms in *Salmonella* Typhimurium, which are related to/based on *E. coli* Rho-Rof, are questionable and need additional data support (see below). Specific concerns are outlined below:

Major Concerns:

1. Rof and Rho are from *E. coli*, However, the authors provided in vivo data from *Salmonella* Typhimurium and proposed many mechanisms about the transcription of *Salmonella* Typhimurium. In Figure S4, the authors showed Rho and Rof are highly conserved across strains and the Rho-Rof interface residues are also highly conserved. However, it is not clear if the structures and sequences of *E. coli* RNAP are highly similar to those of *Salmonella* Typhimurium RNAP. In addition, we don't know if *Salmonella* Typhimurium Rho and Rof would form the same complex as that of *E. coli*. Thus, all the functional conclusions and mechanisms about the transcription of *Salmonella* Typhimurium are unpersuadable. Authors need to provide additional relevant evidence to connect their presumptions on the transcription of *Salmonella* Typhimurium with the determined *E. coli* Rof-Rho complex.

We thank the reviewer for the valuable suggestion. *Salmonella* and *E. coli* are closely-related relatives in the same Enterobacterial family. Both *Salmonella* and *E. coli* encode the highly conserved *rho* and *rof* genes, with >99% identity and 80% identity, respectively. In vitro transcription assay and further structural analysis indicate that *Salmonella* Rof and Rho directly interact and form binary complexes similar to their *E. coli* counterparts.

To address this concern, we have added a new section in the revised manuscript (line 146-160) to introduce the conservation of Rof in bacteria (see the text below). Accordingly, we have included a multiple sequence alignment of Rof proteins in the updated Figure 3, and also included in vitro transcription assays in the Supplementary Figure 5. We believe that these new writing and new data would now provide a better transition from the in vitro analysis to the in vivo functional analyses in *E. coli* and *Salmonella enterica*.

“ As Rho is a highly conserved termination factor, we asked whether Rof is similarly conserved or co-evolved in bacteria. Sequence analysis suggest that *rof* is widely conserved in various bacteria including Enterobacterales, Neisseriales, and Burkholderiales (Supplementary Fig. 6). Within the enterobacterial family which contains many bacterial pathogens, the sequence and structure of Rof are highly conserved and show great similarity to the EcRof (Figure 3a). For example, the gene sequences of Rho, Rof and RNAP in *Salmonella enterica* are very similar to those in *E. coli* (Sequence identity: *rho* 99.5%, *rof* 78.6%, *ropA* 100%, *ropB* 98.7%, *ropC* 98.7%, *ropZ* 100%). It is most likely that Rof regulates Rho-dependent termination in these organisms in a similar manner. Indeed, in vitro transcription assays confirmed that the SeRof could suppress SeRho-mediated transcription termination (Supplementary Fig. 5c). SeRof and EcRof could act interchangeably in vitro. We observed that SeRof inhibit EcRho's termination activity, while EcRof could also inhibit SeRho's termination activity (Supplementary Fig. 5c). Therefore, Rof proteins utilize a conserved antitermination mechanism to suppress Rho-dependent termination. Interestingly, searching the homologs of Rof and Rho in bacterial genomes revealed that Rof is only present in organisms that encode a conserved copy

of rho (Supplementary Fig. 7), suggesting that Rof may have emerged later than Rho to regulate termination in response to certain cues or complex environments.”

Moreover, we have further determined the cryo-EM structure of *SeRof-SeRho* complex (not included in this paper). Although the resolution is low (~6-7 Å), the structure also shows the similar binding conformation between *SeRof* and *SeRho*.

2. “Rof’s N-terminal α -helix undergoes conformational changes upon binding with Rho” Figure S3 shows the superimposition between the Apo-Rof and Rof in this complex, however, it is hard to recognize what the conformational change they refer to. In this reviewer’s opinion, they share similar conformation based on the provided figures. A clearer and more convincing comparison needs to be provided.

We thank the reviewer for the valuable suggestion and regenerate the Figure S3. We address this point in the current version (line 92-95): “The N-terminal α -helix of Rof in our structure displays a ~45° anticlockwise rotation and exhibits large RMSD value (Supplementary Fig. 3a) when compared to EcRof (NMR). While the orientation of the N-terminal α -helix is quite similar to that of VcRof (Supplementary Fig. 3b).”

3. About cryo-EM data: (The overall cryo-EM map quality is high; the density of the interface is clear in the map.)

We thank the reviewer for the favorable assessment.

(1) Figures of model-to-map FSCs need to be present to demonstrate the quality of the models and the corresponding ones (model resolution when FSC=0.5) should also be included in Table 1. The accuracy of the model is important for structural explanations and description.

The figure of model-to-map FSCs has been added in Supplementary Fig. 2f. New Table S1 has been generated with the detail information of model refinements.

(2) In addition, supplementary figures of local density maps with different regions of the models, showing sidechains, are necessary and important to justify the claimed 2.8Å resolution.

We address this point by adding new figure in Supplementary Fig.2h left panel which shows different regions of models (including α -helix and β -sheet) with the density of side chains.

(3) Figures or panels showing high-quality density of interface residues’ side chains should be provided to support the structural observations. Authors could add the map to figure 1 or present it in a supplementary figure.

We address this point by adding new figure in Supplementary Fig.2h right panel showing the density of each key interface residues’ side chain.

(4) Why has the evaluation information of cryo-EM map been deleted from the official validation report?

We add new evaluation files including evaluation information for both cryo-EM map and models.

(5) Table S1 has missed many necessary items, which need to be provided.

New Table S1 has been generated with the all detail information of model refinements and map.

(6) It seems that cryo-EM data was processed using RELION. However, it's puzzling that the authors did not mention and cite RELION.

We cite Zivanov, J. et al. 2018 (ref 50) at the mentioned location in the current version of supplementary text (line 63-64): "Contrast-transfer-function (CTF) estimation was performed using CTFFIND-4.1⁴⁹. Subsequent image processing was performed using Relion 3.0⁵⁰)

Minors:

1. In the abstract: "The structure shows that Rof binds to the open-ring Rho hexamer and inhibits the initiation of Rho-dependent termination" – please rewrite it. The structure doesn't directly show Rof inhibits termination.

We address this point by rewrite it in the current version (line 33-34): "Here we report the cryogenic electron microscopy structure of Rho-Rof antitermination complex. Rof is a conserved antiterminator, it binds to the open-ring Rho hexamer and inhibits the initiation of Rho-dependent termination."

2. Please also provide PDB code for VcRof.

We address this point in the current version (line 84-87): "The densities of Rof in all protomers are clear (we will use the Rof near Rho protomer C and D as an example for the rest part of this paper.) except the one near protomer F, enabling unambiguous rigid-body docking of atomic structure of EcRof (PDB ID: 1SG5). The final models of Rof in the complex exhibit a conserved α - β -sandwich-like structure which was observed in both structures of EcRof and VcRof (PDB ID: 6JIE)." and also point in all the figure legends involving VcRof.

3. It is better to mention in the very beginning of the detailed description of the ρ /Rof interaction interface that you are going to use a particular example between ρ protomers C and D but not at the end of the description.

We thank the reviewer for the valuable suggestion, we address this point in the current version (line 84-85): "The densities of Rof in all protomers are clear (we will use the Rof near Rho protomer C and D as an example for the rest part of this paper.) except the one near protomer F."

4. Figure adjustments:

(1) Potential (my opinion). Figure 1a: The resolution of the cryo-EM map used for this figure seems significantly lower than that provided for the reviewers. Please update the figure with a better representation.

We thank the reviewer for the valuable suggestion, high-resolution map has been updated in Figure 1a.

(2) Figure 1d needs a protomer D label.

Protomer D label has been added in Figure 1d.

(3) Supplementary Figure 1 compares the assembly of ρ /Rof and individual components that may benefit from the addition of a chromatogram with a molecular weight standard.

Molecular weight standard has been added in Supplementary Figure 1.

(4) Supplementary Figure 3A: The choice of colors for two superimposed structures is not clearly distinguishable please choose more contrast (my personal opinion) (yellow and blue for example).

We address this point by regenerate the Supplementary Figure 3a and b. More distinguishable colors have been chosen.

(5)Page 5 line 126. typo Ser8e

The typo has been corrected.

Reviewer #3 (Remarks to the Author):

In the manuscript titled ‘Rof is a key regulator of Rho and transcription termination in bacteria’, Zhang, Zhang, Zhou, and Zhang et al, showed the cryo-EM structure of the Rho-Rof complex and suggested that the Rof inhibits Rho-dependent termination by binding to the PBS (primary RNA binding site) of Rho. Then, the authors switch gears and provide the first biochemical evidence that the Rof regulates the gene expression of virulence factors of Salmonella. Although the individual experiment is valid and provides useful insights into the function of Rof, this reviewer wonders why the authors suddenly shifted the goal of the manuscript from describing the molecular mechanism of anti-termination by Rof using high-resolution structure of E. coli Rho-Rof complex to validating the relation between Rof and virulence factor expression in Salmonella. Addressing both themes is meaningful, but the breadth of research and the depth of discussion on each topic are somewhat lacking.

We thank the reviewer for the favorable assessment.

Major points

1. The title is too broad, providing little information about the manuscript. This reviewer suggests changing the title containing the key idea of the manuscript.

We appreciate the suggestion. We propose to change to the title to: “A widely-conserved protein Rof inhibits transcription termination factor Rho and promotes *Salmonella* virulence program”.

2. On page 5 and Fig 2d, the in vitro transcription assay result is reported, but there is no gel image. This reviewer suggests putting the transcription assay gel image in the supplementary data. This applied to Fig. S10 as well. This is critical because the transcription assay results can be analyzed in any way, and you never

know the result correctly until you see the gel itself. Particularly, previous literature (<https://onlinelibrary.wiley.com/doi/epdf/10.1046/j.1365-2958.1998.00981.x>) reported that the Rof function could not be confirmed *in vitro*.

We thank the reviewer for the valuable suggestion. Due to the limiting availability and strict regulation of P³² radioactive material, we are not able to perform the transcription assay with ‘hot’ sequencing gels in the lab. Alternatively, we opted to the fluorescence-based *in vitro* transcription assay to measure Rho-dependent termination efficiency and Rof’s anti-termination activity. The fluorescence-based assay has been previously reported and successfully used in a number of published works (<https://www.nature.com/articles/s41467-022-31871-7>, <https://www.nature.com/articles/s41589-020-00653-x>).

Briefly, in the the fluorescence-based *in vitro* transcription assay, the small-molecule fluorophore Thiazole Orange (TO)-Biotin could fluoresce upon binding to Mango-III RNA sequence during transcription. The DNA template pT7A1- λ cro contains a T7A1 promoter, a region of the λ chromosome encompassing the cro ORF, and a Rho-dependent tR1 terminator, followed by the Mango-III encoding sequence. The Run-off transcripts with the Mango RNA could be detected as high fluorescence signal, whereas the Rho-terminated RNA products without RNA Mango sequence has low fluorescence.

3. In the Fig 3b (and Fig S6a), the colony formation of Salmonella Typhymurium was normal in Δ rof + pRof but lost in Δ rof + pL-Rof. This reviewer could not find what pRof is. In addition, main text writes “overexpression of Rof inhibited E. coli growth under osmotic stress (Fig. 3a) and completely inhibited Salmonella growth and colony formation in standard LB medium (Fig. 3b, Supplementary Fig 6).”, but the graph in Fig3b seems to suggest high salt does not affect the Salmonella cell growth both in WT and rof. In addition, the figure legend does not comment on this graph.

We apologize for the confusion. pRof is a rescue/complementation plasmid, in which the constitutive pL promoter on the pZE12 vector was replaced with the own promoter of *rof*, so that Rof was not constitutively overexpressed to avoid its lethal effect). We have added this important information to the figure legends as well as the material and methods.

The original Figure 3b (the right panel) showed that *rof* null mutant did not show growth defect under high salt condition. To avoid confusion and better clarity, we have removed the old data in Figure 3b in question, and replaced it with a more informative growth curve from the original Supplementary Figure 6. The new Figure 3c now clearly shows the growth defect of Rof overexpression mutant, as well as the absence of growth alteration for the *rof* null mutant.

4. The authors claim that the results show that the Rof regulates virulence gene expression, in particular, SPI-1 genes for the first time. To solidify the function of the Rof in pathogenesis, this reviewer suggests the authors comment on the upstream pathway – for example, how the Rof is turned on (transcriptional regulation?).

Thanks for the interesting question. The SPI-1 genes are under complex regulations in *Salmonella*, which remains a central topic of research in the field. Dozens of transcription regulators including several different two-component systems, responding to various environmental signals, have been reported to regulate the

expression of SPI-1 genes, such as EnvZ/OmpR, SirA/BarA, PhoP/Q, RcsB/C, QseB/C, Fnr, Crp, Lrp, to name just a few. Likewise, it is not immediately clear which upstream pathway(s) activate/repress the expression of rof. Using a transcriptional GFP reporter fused to the rof promoter, a preliminary experiment showed that high-salt may be an activating signal of Rof (see the figure below), providing some hints of the upstream regulators/pathways. To adequately address this important question, we are currently following up on these observations and aim to delineate the regulatory cascade of Rof in a follow-up paper.

5. This reviewer believes that the first part of the manuscript, concerning the Rho-Rof complex structure, could benefit from a more exploration of different perspectives. For example, did Rof induces any conformational change in Rho? If so, how would that affect the termination?

We appreciate the suggestion. We address this point in the current version (line 231-238): "Interestingly, it has been previously reported that the PBS ligand, ATP, and NusG would trigger Rho ring closure and the ring-closure is the crucial step for Rho-dependent transcription termination as “catch-up” model⁴⁶. Our Rof-Rho complex shows that Rho is in a ring-open state, and following submission of our manuscript, preprint reports Rof could conformationally regulate the Rho ring dynamics and induce a hybrid conformation⁴⁷. Together with our results that NusG could partially restore the Rho’s termination activity which inhibited by Rof, it may suggest the NusG would trigger the Rho ring closure even in the presence of Rof. Further structural analysis of the TEC-Rho-NusG pre-termination complex in the presence of Rof would provide greater clarity.”

Minor points

1. Line numbers in the manuscript are necessary for referencing specific content within the text.

We thank the reviewer for the valuable suggestion, line numbers have been added in the current version.

2. On page 3-4, “High-resolution data clearly show additional Rof density near the N-terminal of Rho with local resolution at ~4-5 Å (Fig. 1a and Supplementary Fig. 2). The densities of Rof in all protomers are clear except the one near protomer F, enabling unambiguous rigid-body docking of atomic structure of EcRof (PDB ID: 1SG5).”. In this reviewer’s observation, the densities of the two (not only one) Rof protomers on the Rho protomers beside the opening are less clear than the other four, and as the authors commented one is much worse than the other. Do you think this is due to the occupancy or flexibility of the molecules? Why?

We agree with the reviewer that the poor density of Rof near protomer F is due to the flexibility of the molecule as losing parts of protein-protein interactions with Rho. We carefully compared the two Rof on the Rho protomer beside the opening, and believe the density of Rof near Rho protomer F is much worse than other five. We address this point in the current version (line 111-114):“It appears that the open-ring gap between protomer F and A is causing a disruption in the interactions between Rof’s β 3- β 4 loops and Rho. As a result, the density of Rof in protomer F is considerably worse and has lower local resolution compared to the other protomers of Rho.”

3. On page 7, “This suggests that Hfq might be a structural homolog of Rof acting similarly on Rho, despite the low similarity in primary sequence (Supplementary Fig. 9).”. However, in Fig S9, the sequence homology looks pretty high. Could you provide the exact percentage of the sequence homology?

We address this point in the current version (line 215-217): "This suggests that Hfq might be a structural homolog of Rof acting similarly on Rho, despite the low similarity in primary sequence (Sequence identity between EcRof and EcHfq, 4.9%, Supplementary Fig. 8).”

4. In discussion, “Based on these results, we proposed that the initiation of Rho-dependent termination has two distinct mechanisms: one is PBS ligand-dependent initiation which could be regulated by Rof, and the other is PBS ligand-independent initiation which is mediated by NusG.” Do the authors think these relate to ‘ catch-up (or RNA-dependent) termination ’ and ‘ stand-by (or RNAP-dependent) termination ’ (<https://doi.org/10.1038/s41467-022-29321-5>)?

We thank the reviewer for the valuable suggestion, the paper about the two model is very important and has been cited in current version (line 231-238):“Interestingly, it has been previously reported that the PBS ligand, ATP, and NusG would trigger Rho ring closure and the ring-closure is the crucial step for Rho-dependent transcription termination as “catch-up” model⁴⁶. Our Rof-Rho complex shows that Rho is in a ring-open state, and following submission of our manuscript, preprint reports Rof could conformationally regulate the Rho ring dynamics and induce a hybrid conformation⁴⁷. Together with our results that NusG could partially restore the Rho’s termination activity which inhibited by Rof, it may suggest the NusG would trigger the Rho ring closure even in the presence of Rof. Further structural analysis of the TEC-Rho-NusG pre-termination complex in the presence of Rof would be needed to make it more clear. ”

For the “stand-by” model, we think it is not clear whether NusG is needed in this model as only NusG N-terminal were found in the structure of “stand-by” model, and the NusG C-terminal would crash with Rho according to the structures. It would be great interesting to study the function of NusG in “stand-by” model using *in vitro* system, but may not discussed in this paper due to the space limitation.

5. The Rof forms a stable complex with rho hexamer regardless of the RNA ligand. Then, how would Rof differentiate the virulence-related genes from other housekeeping genes? If this manuscript is the first report showing that the Rof regulates virulence-related gene expression, this might need to be suggested at least in the Discussion.

Our discovery is based on the previous study that shows that the SPI-1 virulence genes are regulated by Rho via an intermediate regulator H-NS, which binds to AT-rich sequence and silences pervasive transcription. Therefore, Rof adds an upstream regulatory element in the existing model containing Rho and H-NS. As suggested by this reviewer, we have included a brief description of the existing model in the first paragraph of the Discussion in the revised manuscript (line 202-203).

“Under OFF conditions, the pervasive transcription of SPI-1 genes (which contains AT-rich sequences) is silenced by both chromatin-binding protein H-NS and Rho.”

6. The structure of E.coli Rof is solved, but only Salmonella Rof function is described. Could you (at least briefly) suggest the physiological role of Rof in E.coli?

Thanks for this interesting question. Our functional study was mainly performed in *Salmonella* using deletion, complementation, and over-expressions, leading to the discovery that Rof is a functional regulator under high-salt and low oxygen conditions. Therefore, it's tempting to speculate that Rof also regulates adaptation to high salt conditions in *E. coli*. This idea received support from our overexpression experiments in *E. coli*. Overexpression of Rof showed a strong impact on growth under high-salt conditions in *E. coli* (originally in Figure 3a, now in Figure 3b). We have now included this idea in the revised manuscript (line 162-164).

“Interestingly, our analysis revealed a regulatory role of Rof in stress responses in *E. coli* and *Salmonella enterica*, two similar and well-established model organisms to study bacterial physiology and pathogenesis, respectively.”

7. Page 3, previous reported  previously reported?

The typo has been corrected.

8. In Fig. S3, How about drawing a figure showing the RMSD values of Ca in Rof. Either the color or thickness of the putty can describe the large RMSD values (<https://pymolwiki.org/index.php/ColorByRMSD> or https://pymolwiki.org/index.php/File:B_factor_putty.png). If one Rof in the protomer F deviates from others, draw the one and compare it with the other five might show it more clearly.

We thank the reviewer for the valuable suggestion, we address this point by adding ColorByRMSD model in Supplementary Fig. 3.

REVIEWERS' COMMENTS

Reviewer #1 (Remarks to the Author):

The authors have adequately addressed my concerns in their revised manuscript.

Reviewer #2 (Remarks to the Author):

The whole quality has been largely improved. New supportive data have been supplemented in the revised manuscript, including additional in vivo and in vitro data and more structural evaluations. All my concerns have been addressed. It would enhance the manuscript if the authors could include 'sequence alignment and statistical analysis' details in the 'Methods' section. Some minor typos remain:

Fig. S1. Line 202. 'Molecualr' to 'Molecular'.

Fig. S9. Line 276. 'it's mutation' to 'its mutation'.

Reviewer #3 (Remarks to the Author):

The authors addressed most of my previous comments and incorporated revisions into the manuscript. While the section on the role of Rof in stress environments appears to be improved or better organized, the structural insights from the Rof-Rho complex regarding the molecular mechanism of Rof remain relatively straightforward. For the revised version, I have only a few minor points to address:

Minor points

Lines #103-104: hydrogen bonds  hydrogen bonds and salt bridges

Line #213: basepairing  base-pairing

Fig3b, c legend: the information on plasmids used in Fig. 3b is described in Fig. 3c. The information might need to be written in Fig.3b legend.

REVIEWERS' COMMENTS

Reviewer #1 (Remarks to the Author):

The authors have adequately addressed my concerns in their revised manuscript.

We thank the reviewer for the favorable assessment.

Reviewer #2 (Remarks to the Author):

The whole quality has been largely improved. New supportive data have been supplemented in the revised manuscript, including additional in vivo and in vitro data and more structural evaluations. All my concerns have been addressed.

We thank the reviewer for the favorable assessment.

It would enhance the manuscript if the authors could include ‘sequence alignment and statistical analysis’ details in the 'Methods' section.

“Sequence alignment and statistical analysis” has been added in “Methods” section.

Some minor typos remain:

Fig. S1. Line 202. ‘Molecular’ to ‘Molecular’ .

Fig. S9. Line 276. ‘it’ s mutation’ to ‘its mutation’ .

The typo has been corrected.

Reviewer #3 (Remarks to the Author):

The authors addressed most of my previous comments and incorporated revisions into the manuscript. While the section on the role of Rof in stress environments appears to be improved or better organized, the structural insights from the Rof-Rho complex regarding the molecular mechanism of Rof remain relatively straightforward.

We thank the reviewer for the favorable assessment.

For the revised version, I have only a few minor points to address:

Minor points

Lines #103-104: hydrogen bonds  hydrogen bonds and salt bridges

The typo has been corrected.

Line #213: basepairing  base-pairing

The typo has been corrected.

Fig3b, c legend: the information on plasmids used in Fig. 3b is described in Fig. 3c. The information might need to be written in Fig.3b legend.

Fig3b and Fig3c have used different plasmids. The relevant information on all plasmids have been included in the legend.